# Novel evidence that elk were historically native to the Sierra Nevada, and recent range expansions into the region

Richard B. Lanman[1]*, Thomas J. Batter[2], Cody J. Mckee[3]

1 Institute for Historical Ecology, Los Altos, California, United States of America, 2 Elk and Pronghorn Program, Big Game Unit, Wildlife Branch, California Department of Fish and Wildlife, West Sacramento, California, United States of America, 3 Elk and Moose Program, Game Division, Nevada Department of Wildlife, Reno, Nevada, United States of America

☯ These authors contributed equally to this work.
¤ Current address: Bighorn Sheep Program, Wildlife Management Division, New Mexico Department of Game and Fish, Santa Fe, NM, United States of America
* ricklanman@gmail.com

**Data Availability Statement:** All relevant data are within the manuscript.

**Funding:** The author(s) received no specific funding for this work.

## Abstract

Elk (*Cervus canadensis*) have been considered non-native to the Sierra Nevada mountain range of California and Nevada. However, elk have steadily increased their range southward from the Cascade Range into the northern Sierra Nevada over the last century. Recent reports also reveal Rocky Mountain elk moving northwards into the southern Sierra Nevada. Dispersals of lone bull elk from 2019–2022 have occurred to the central Sierra Nevada south of Lake Tahoe. These recent range expansions of elk herds and long-distance dispersals of individual elk raised questions about the possible historical presence of elk throughout this mountain range. Herein we conducted a broad investigation into historical newspaper accounts and other early explorer and naturalist observer records, museum specimens, Late Holocene zooarchaeological records, and indirect evidence including toponomastic references and Native American ethnographic and ethnolinguistic information. Taken in total, a variety of data sources suggest elk inhabited portions of the Sierra Nevada and the adjacent northwest Great Basin from the Late Holocene through historical times. Positive records were not numerous, suggesting that historically elk were not abundant, and nearly extirpated during the California Fur Rush of the early nineteenth century.

## Introduction

The California Department of Fish and Wildlife (CDFW) 2018 Elk Conservation and Management Plan excluded the Sierra Nevada of eastern California and western Nevada from their map of historical elk (*Cervus canadensis*) range (Fig 1) [1]. This estimate persists in recent authoritative texts [2, 3] but was based on twentieth century accounts written after possible extirpation of elk in the region in the nineteenth century [4, 5]. The nearest elk historically were thought to be Rocky Mountain elk (*C. c. nelsoni*) located in northeast California's

**Competing interests:** The authors have declared that no competing interests exist.

**Fig 1. Estimated historical distribution of California's three elk subspecies.** Rocky Mountain elk (yellow) shown in northeastern California north of the Sierra Nevada, Roosevelt elk (blue) in northwestern California, and tule elk (pink) in California's Central Valley and central coast regions. As shown, the Sierra Nevada were excluded from historical elk range. Reprinted from CDFW Elk Conservation and Management Plan under a CC BY license, with permission from CDFW, original copyright 2018.

southern Cascade Range and tule elk (*C. c. nannodes*) relegated to "the Central Valley and the grasslands and woodlands of central California's Coast Range" (Fig 1). Tule elk have a well-established habitat preference for more open terrain and generally avoid forested habitat and higher elevations characteristic of the Sierra Nevada [5]. This strong habitat preference is likely related to physical and behavioral adaptations for drier, Mediterranean-climate conditions [6, 7]. In describing primitive North American elk distribution in 1951, O. J. Murie cited a lack of available records of elk in easternmost and southernmost California and suggested that elk did not occupy Nevada "in any considerable numbers". Murie recognized, however, that elk have occurred beyond the boundaries presented and a more extensive record would likely

reveal a greater historical distribution [4]. In 1969 D. R. McCullough identified reliable historical records and museum specimens consistent with Rocky Mountain elk historical presence in Shasta and Siskiyou Counties of northeastern California [5, 8, 9]. Until then CDFW had disputed that this subspecies had existed in California, aligning with Murie's perspective [4, 10]. However, McCullough's findings were subsequently accepted by CDFW in their 2018 elk plan [1].

In partial agreement with Murie's work, the 1997 Nevada Department of Wildlife (NDOW) Elk Species Management Plan established that elk were native to Nevada, but densities were low wherever they occurred [11]. In contrast to the CDFW Elk Management Plan, the NDOW Elk Species Management Plan noted that "Newspaper accounts report hunter kills at Lake Tahoe and Honey Lake Valley. . ." [11]. These two accounts, both in the Sierra Nevada region near the California and Nevada border, were based on an extensive search of historical newspaper and other observer records, but the specific citations were not listed. Nevertheless, the NDOW findings challenged prevailing thinking that elk never occupied the Sierra Nevada region.

Few contemporary studies are conducted with their primary goal the determination of the historical range of a given species [12]. Understanding ecosystem reference conditions in California and Nevada present challenges due to the relatively late establishment of zoological collections in museums in these states until the late nineteenth and early twentieth century [13]. The California Fur Rush of the first half of the nineteenth century began as fur trappers and hunters ventured into the Sierra Nevada region beginning with Jedediah Smith in 1827 and Peter Skene Ogden in 1828 [13–15]. These early venturers may have rapidly depleted elk in the Sierra Nevada before naturalists arrived to document their presence. For example, after killing over 130 elk in a single winter season at the mouth of the Columbia River from November 1805 to March 1806, Lewis and Clark lamented how difficult it had become to hunt elk without traveling long distances from their camp [16]. These factors may have led to similar underestimation of the historical ranges of elk and other mammalian species in California. In fact, several recent historical ecology studies have expanded the accepted historical range of species long thought to be non-native to the Sierra Nevada region, including physical evidence of the historical presence of North American beaver (Castor canadensis) in the high Sierra Nevada [13, 17] and the historical nativity of the California red-legged frog (*Rana draytonii*) to the Yosemite region [18]. Although direct evidence of historical presence of the gray wolf (*Canis lupus*) in the Sierra Nevada has not been found (although one museum specimen was collected near Litchfield, Lassen County, California just north of the Susan River) [19, 20], wolves have recently colonized both the northern and central Sierra Nevada, with packs currently (re-) established in Plumas and Tulare Counties, respectively–demonstrating that at least portions of the Sierra Nevada are suitable wolf habitat [21]. Similarly, a historical ecology study of the presence of wolverine (*Gulo gulo*) found historical observer records only in the southern subalpine and alpine Sierra Nevada [22], but recent discoveries of wolverine dispersers to the Lake Tahoe and Yosemite National Park regions in 2008 and 2023 raise questions about the former distribution of this carnivore in the Sierra Nevada [23].

The Sierra Nevada is dominated by coniferous forest, a habitat type often associated with Rocky Mountain elk, and there are no obvious biogeographical barriers that would prevent Rocky Mountain elk range expansion south from the southern Cascade Range to the northern Sierra Nevada. The uncited historical newspaper accounts mentioned above and recent studies expanding the historical range of other mammals to the Sierra Nevada, prompted our search for historical evidence of elk in the region. To overcome the paucity of evidence typical of historical ecology studies [24], we evaluated multiple lines of evidence, similar to a recent investigation for evidence of late Holocene and historical records of bison (*Bison bison*) in northeastern California and northwestern Nevada [25]. Our study goals were to research and

unify broad data sources for historical and Late Holocene evidence of elk in the Sierra Nevada, and to update information on the region's current (re)colonization by elk.

## Methods

### Study areas and time period

The Sierra Nevada mountain range is 640 km (400 miles) long on its north-south axis, and 80–130km (50–80 miles) wide east–west. Its northern border stretches from the North Fork Feather River east to Fredonyer Pass south of Mount Lassen, then further east along the Susan River to its terminus at Honey Lake, California. Its southern border is at Tehachapi Pass where the Sierra Nevada give way to the Tehachapi Mountains to the southwest [13]. The Sierra Nevada's peaks steadily rise in elevation from 2,130 m in the north to 4,413 m tall Mount Whitney in the south. With the exception of zooarchaeological specimens dating to the Subatlantic age (< 2,500 years BP) of the Late or Meghalayan Holocene [26], the time period investigated for historical records of elk in the region begins with the arrival of Spanish explorers to California in 1769 [27].

### Specimen and records search

First, we searched museum records and zooarchaeological specimens for physical evidence of elk's historical presence in the Sierra Nevada region. For museum specimens we examined the Arctos Multi-Institution and Multi-Collection Museum Database (Arctos) (https://arctos. database.museum/), the Mammal Networked Information System (MaNIS) (http://manisnet. org/), and the Integrated Digitized Biocollections (iDigBio) (https://www.idigbio.org) via Boolean searches using the terms "elk", "wapiti", "Cervus canadensis", and "Cervus elaphus". In addition, we contacted curators of mammal collections at the California Academy of Sciences (CAS), the Berkeley Museum of Vertebrate Zoology (MVZ), and the National Museum of Natural History (USNM) for elk specimens collected in the Sierra Nevada region. For zooarchaeological records of elk in the region we queried FAUNMAP (http://www.ucmp. berkeley.edu/neomap/search.html) for "Cervus elaphus" and "Cervus canadensis" remains and references listed in recent reviews of large ungulates (bison and elk) in the Great Basin [25, 28].

We used Web of Knowledge and Google Scholar to search for historical observer records by the earliest explorers and naturalists to the Sierra Nevada as well as those in citations in publications that reviewed the historical ranges of other California mammals in the region [13, 20, 22, 29–31].

Contemporary observer records were obtained by CDFW and NDOW biologists. No permits were required for the described study, which complied with all relevant regulations.

### Indirect evidence search

Indirect evidence searches included searches for geographic place names using the United States Geological Survey (USGS) Geographic Names Information System (GNIS) (https:// geonames.usgs.gov/pls/gnispublic) and toponomastic references for California, Nevada, and the Lake Tahoe area [32–34]. Because ethnographic and ethnolinguistic information does not provide a specific written historical record, we considered it indirect evidence. We researched Native American ethnographic and ethnolinguistic information and geographic place names including the word "elk" as referenced below.

## Results

### Historical newspaper accounts and other observer records

First, we located the two previously uncited historical newspaper references in the NDOW report, which described Sierra Nevada elk hunts in 1867 and 1868, respectively. The first historical newspaper account was printed in The Carson Daily Appeal in 1867 [35] and reads as follows:

> *"ELK AT LAKE BIGLER–We learn that two elk were killed at Lake Bigler, beyond the Zephyr Cove House, during last week. One of them weighed about five hundred pounds."*
>
> *- The Carson City Daily Appeal 1867.*

Lake Bigler was the historical name of Lake Tahoe, named for California's third governor, John Bigler. However, because Bigler was a secessionist, the United States Department of the Interior renamed it Lake Tahoe during the Civil War [33]. Zephyr Cove House was established by Andrew Gardenier, who homesteaded 65 hectares (160 acres) and built his inn in 1862 on Lake Tahoe's southeastern shore to cater to silver miners. It was located at Zephyr Cove, Douglas County, Nevada [36]. The second elk hunt account was described in a pair of newspaper references in 1868 in the vicinity of Honey Lake Valley, Lassen County, California:

> *"RETURN OF THE HUNTERS–About two weeks since Alderman Dimock and one or two other gentlemen left this city* [Virginia City] *for a hunt in the direction of Honey Lake. . . Night before last they returned to this city, bringing several trophies of their prowess. During their excursion they killed an elk, weighting 404 pounds, and a number of deer, antelope, and smaller game."*
>
> *- Territorial Enterprise 1868.*

This account was printed in the Territorial Enterprise (Virginia City) in 1868 [37] and accompanied by a second report in the Virginia City Daily Trespass [38]: "*a hunting excursion to Honey Lake Valley. . . to Doyle's ranch, in Honey Lake Valley, which place they made their headquarters for a seven-days hunt. . . Hunting was done principally on horseback. For the first three days all the game killed was a noble elk, which weighed 404 pounds, having antlers at least four feet in length.*"
- The Daily Trespass 1868.

Doyle's Ranch still exists and is located in Milford, Lassen County, California on the southwest shore of Honey Lake, the terminus of the Susan River. Honey Lake Valley lies south of Honey Lake, 107 km north of Lake Tahoe. Both Honey Lake Valley and Lake Tahoe are south of the Susan River, placing both newspaper accounts in the Sierra Nevada (Fig 2) [39]. The two newspaper accounts are 122 km and 260 km south, respectively of CDFW recognized historical elk range (Modoc County's southern border) [1].

In addition to the two newspaper accounts of elk hunts mentioned in the 1998 NDOW Elk Management Plan at Zephyr Cove, Nevada and Milford (Honey Lake Valley), California, respectively, we located four additional historical elk observer records in the Sierra Nevada, bringing the total to six locations (Table 1).

The earliest record was in 1847 by Chester Ingersoll, who meticulously described his wagon train journey over the Sierra Nevada crest and down along the South Fork Yuba River then to Bear River Valley, Placer County, CA where he reported "*Game plenty, bear, deer and elk. . .*" [40]. Then, in 1874, two different elk hunts in Honey Lake Valley and Round Valley, Plumas

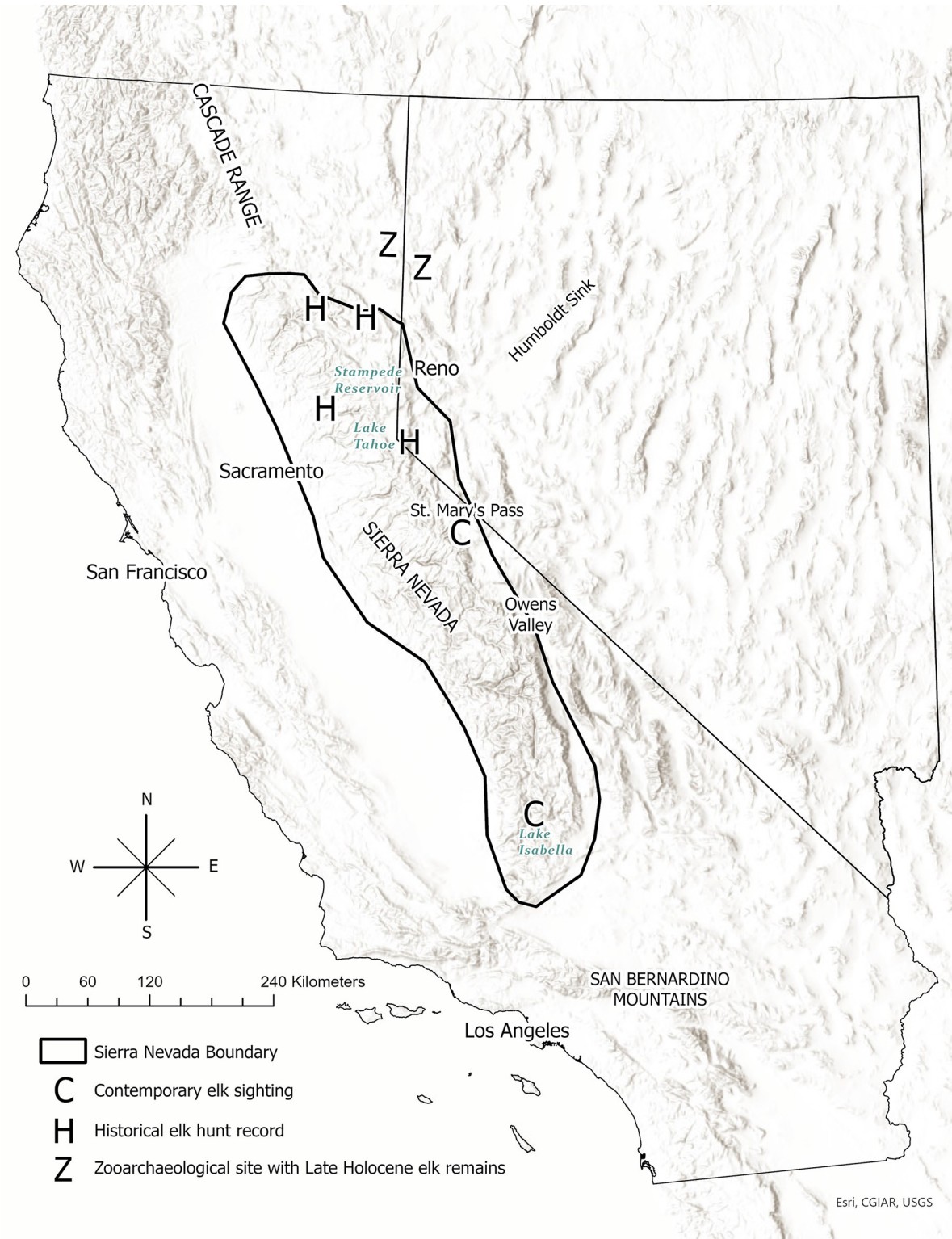

**Fig 2. Map depicting locations of six nineteenth century elk observer records at four locations "H": Honey Lake, Lake Tahoe, Round Valley, and the Bear River canyon in the Sierra Nevada.** Solid line depicts the boundaries of the Sierra Nevada in California and Nevada [39]. Two Late Holocene zooarchaeological records of elk "Z" are located in the extreme northwest Great Basin, just north of the Sierra Nevada. Habitat suitability is reflected by the contemporary presence "C" of breeding elk today in the northern Sierra Nevada south to Stampede Reservoir and the southern Sierra Nevada north to Lake Isabella. The dispersal of a bull elk to St. Mary's Pass "C" south of Lake Tahoe is also noted. Images in map collected from public domain open sources (Esri, CGIAR, USGS).

Table 1. Summary of historical observer records of elk hunts in the Sierra Nevada.

| Record Date | Historical Elk Hunt or Observer Record Location | Approximate GPS Location (DMS) | Elevation | Reference |
|---|---|---|---|---|
| Sep 23, 1847 | Bear River valley 1.6 km north of Dutch Flat, Placer County, CA | 39˚ 13' 26" N, 120˚ 49' 40" W | 853 m (2,800 ft) | [40] |
| Oct 17, 1867 | Zephyr Cove, Douglas County, NV | 39˚ 00' 27" N, 119˚ 56' 55" W | 1,900 m (6,234 ft) | [35] |
| Feb 11–12, 1868 | Milford (Honey Lake Valley), Lassen County, CA | 40˚ 10' 11" N, 120˚ 22' 10" W | 1,287 m (4,222 ft) | [37, 38] |
| Dec 4, 1874 | Honey Lake Valley, Lassen County, CA | 40˚ 14' 41" N, 120˚ 18' 12" W | 1,217 m (3,993 ft) | [41] |
| Dec 23, 1874 | Round Valley, Plumas County, CA 12 km east/southeast of Lake Almanor | 40˚ 6' 57" N, 120˚ 58' 04" W | 1,378 m (4,521 ft) | [42] |
| Jan 8–9, 1875 | Honey Lake Valley, Lassen County, CA | 40˚ 14' 41" N, 120˚ 18' 12" W | 1,217 m (3,993 ft) | [43, 44] |

County reported 15 and 13 elk taken, respectively [41, 42]. The last elk hunt newspaper record was described in two newspapers on Jan 8 and 9, 1875 where a dozen elk were taken in the vicinity of Honey Lake and described thusly: "*Judging from the size of their bodies, they must have been monsters when living. In order to save a few dollars in freight the heads and legs were cut off of all the animals.*" [43, 44].

After reviewing historical records for elk in the Sierra Nevada, we searched for other records in nearby regions that appeared reliable because they mentioned both deer and elk in the same observation. In the Great Basin adjacent to the Sierra Nevada's eastern border, a report of elk in the Humboldt Sink was given by Thomas Jefferson Farnham, who explored California and Nevada in 1840: "*On the northwest side of this Desert* [Great Basin] *is a partially fruitful region, called the Vale of Mary's River* [Humboldt River]... *There are, however, some pretty groves of aspen and pine to be found along the stream and in the hills, among which live a few red deer and elk.*" [30]. Farnham also noted the occurrence of elk south of the Sierra Nevada in California. In 1840 he described the course of the Mojave River as it approached its headwaters in the San Bernardino Mountains, as "*gurgling through narrow vales covered with grass and fields and forests in which live the deer, the black bear, the elk, the hare, and many a singing bird*" [30]. The Mojave River mainstem begins in the San Bernardino Mountains, 128 km southwest of Tehachapi Pass, the southern border of the Sierra Nevada. John Boyden "Grizzly" Adams, resided at Long Barn in the Sierra Nevada, near Sonora Pass in Tuolumne County, and described hunting elk in the 1850's but these accounts were not clearly in the mountains versus the Central Valley of California and further may have been sensationalized for the San Francisco newspapers [45].

In contrast, Jedediah Smith, the first Anglo-American to cross the Sierra Nevada in 1827 at Ebbett's Pass, 50 km south of Lake Tahoe, did not describe seeing elk [46]. Peter Skene Ogden, the discoverer of what is now the Humboldt River in 1828 and who explored it in 1829, also did not mention elk on his return from the Humboldt Sink over the southern Cascades to Oregon [47]. In the diary of Charles Preuss, who along with John Fremont were the first Anglo-Americans to see Lake Tahoe in early 1844 on Fremont's second expedition, no elk observations were reported [33].

## Museum records and Late Holocene zooarchaeological specimens

There were no museum specimens found for *Cervus canadensis* or *Cervus elaphus* from the Sierra Nevada region of either California or Nevada utilizing Boolean searches of the ARCTOS, MaNIS, or iDigBio museum databases, nor did the curators of mammalogy collections we contacted have specimens that may not have been digitized or made accessible online.

Although noting that the NDOW 1997 Elk Species Management Plan did not provide publication details for the two historical newspaper accounts, it cited a comprehensive review of archaeological sites in the northwestern Great Basin which found elk remains at two Late

Holocene sites: CA-LAS-206/15 located in Secret Valley, Lassen County, California, 34 km north of Honey Lake Valley, and 26WA5649 at Dry Valley, Washoe County, Nevada, 35 km east of Honey Lake Valley (11,28) (Fig 2) [28].

## Indirect evidence of elk in the Sierra Nevada

There are numerous place names including the word "elk" in California and Nevada, but only a single occurrence in the Sierra Nevada counties of these two states, an Elk Point in Douglas County, Nevada on Lake Tahoe [33, 48]. Coincidentally, Elk Point is only 2.0 km south of Zephyr Cove, although its etymology is unknown [33]. Searches of other toponomastic references yielded no further elk place names in the Sierra Nevada [32, 34].

In support of the historical newspaper and observer accounts, there is ethnographic support for elk in the Sierra Nevada. The Washoe People had a word for elk, *hañakmuwe* [49]. The Northern Paiute had a word for elk, but this does not provide added support since the Northern Paiute territory included portions of Oregon and Idaho where elk were historically present. However, there is a specific reference to the Honey Lake Valley Paiute, which were local to the Honey Lake Valley, stating that elk were "among the large animals captured" [50]. Similarly, the Mountain Maidu's territory likely extended at least to the western edge of the Honey Lake Valley, and although the other side of their territory extended to the edge of the Central Valley, a fairly specific reference to elk as a food item in the mountainous region states: "In the mountains, deer, elk, mountain-sheep, and bear were plenty; while in the Sacramento Valley there were great herds of antelope" [51]. There is an Owens Valley Paiute legend of *H'ai'nanū* who travels with his brother to Convict Lake then Mammoth in Mono County, California then over a ridge and shoots *soikwoi* or elk [52]. Lastly, a historical newspaper account of an interview with Northern Paiute Jonathan Sides in the Reno Gazette-Journal in 1880: *"It was interesting to hear to this cultivated Piute* [sic] *talk of the early days in Nevada, before the white man's foot had been set upon its soil. Deer, he said, are not so plenty as they used to be. There were, in the old time, lots of elk, antelope and mountain sheep in the Sink of the Humboldt. . .Now the wild sheep, the elk and the antelope are never seen."* [53].

## Reports of recent elk range expansions into the Sierra Nevada

The CDFW Elk Management Plan's current elk range maps indicate that by 2017, natural range expansion of Rocky Mountain elk had occurred southwards from Modoc County through Lassen, Plumas, and Sierra Counties [1]. Elk calving grounds were identified in a 2020 study of the Plumas National Forest, which extends south to northern Sierra County, and reported that breeding elk herds had been established there by the early 2000s [54]. Presently, three spatially distinct herds composed of both sexes and varying age classes occupy Plumas and Sierra counties in the northern Sierra Nevada, near Humbug Valley and Lake Davis, Plumas County, and Stampede Reservoir, Sierra County, the latter 30 km north of Lake Tahoe (CDFW internal data). Recently, a genetics study of California's elk populations found that the elk in the northern Sierra Nevada are intergrade Roosevelt elk x Rocky Mountain elk subspecies [55]. Abundance and space use of these populations are not well understood, however elk occupation and reproduction in these regions indicate that habitat is suitable for elk, in at least the northern Sierra Nevada.

Although the 2018 CDFW Elk Management Plan showed no elk in the southern Sierra Nevada, wildlife biologists since 2021 have observed elk, including a substantiated photograph of at least three cows, with darker pelage consistent with Rocky Mountain elk, in the Sequoia National Forest/Lake Isabella area of Kern County at elevations above 800 m (A. Gwinn, CDFW and N. Kelly, USDA Forest Service, personal communication). These elk are thought

to be dispersers from the Tehachapi herd, whose founders were Rocky Mountain elk translocated from Yellowstone National Park to the ranch of Rex Ellsworth next to Fort Tejon in Kern County's Tehachapi Mountains in 1967 [1]. These conclusions are supported by three recent documented elk-vehicle collisions on Highway 58, near Tehachapi Pass, the boundary between the Tehachapi Mountains and the southern Sierra Nevada (A. Gwinn, CDFW and N. Kelly, USDA Forest Service, personal communication). There is no evidence that the tule elk removed from Yosemite Valley in 1933 and transported to the Owens Valley in Inyo County on the southeastern edge of the Sierra Nevada [56] have expanded their range westward, as is expected due to their high fidelity for more open range and lower elevations.

### Recent long-distance dispersals of Rocky Mountain elk in the Sierra Nevada

There are numerous recent reports of elk making long-distance dispersals southwards through the Sierra Nevada. In 2019, a radio collared elk cow left the Alturas area in Modoc County and was tracked dispersing south until it was hit on U.S. 395 just north of Reno, an over 200 km linear distance (CDFW internal data). In addition, a young bull elk was filmed on a road near Weimar, Placer County in August 2022, 70 km due west of Lake Tahoe (S. Holm, CDFW, personal communication). An unusually long-distance dispersal event was detected in 2019 and 2020, when a bull elk was photographed on trail camera in El Dorado County near Lake Tahoe and then redetected at Saint Mary's Pass (just northwest of Sonora Pass) in Alpine County, a total of 335 km south of Modoc County and 70 km further south than Lake Tahoe, [57] (Fig 2). Analysis of fecal DNA collected near Lake Tahoe, Saint Mary's Pass, and intervening sites confirmed this journey was made by the same individual bull elk. In October 2022, a bull elk was observed by CDFW staff ~10 km south of Bridgeport in Mono County, east of US Route 395 and south of State Highway 270, a distance of 110 km south by southeast of Lake Tahoe (D. Taylor, CDFW, personal communication). In September 2023, a bull elk was again observed near Bridgeport comingling with a flock of sheep (D. Taylor, CDFW, personal communication). It is unknown if these latter observations are of distinct individuals, or the same individual detected in Alpine County in 2020. There were also dispersals to the northwest Great Basin in 2023 with observations of dispersing bulls on Peavine Mountain bordering the north side of Reno and Hallelujah Junction just inside the California Border on US 395, 34 km northwest of Reno. Presumably, these were dispersers from one of the 3 established herds in Plumas or Sierra Counties (C. McKee, NDOW, personal communication).

### Discussion

Multiple direct and indirect data sources indicated that elk were historically present in the northern Sierra Nevada. Three of the six historical observer records establish the presence of elk with reasonable specificity, and two more of the six report elk in the same locations as the initial three suggesting they are *bona fide*. The 1867 Zephyr Cove, Lake Tahoe account of a "500 pound" (227 kg) elk would not be a mule deer (*Odocoileus hemionus hemionus*), where adult bucks average 74 kg (163 lbs) dressed and about 25% heavier or 106 kg (233 lbs) live [58, 59]. Although mule deer size is quite variable, the largest trophy animal taken in northeastern California had a live weight of 181 kg (400 lbs) [10]. The 1868 Honey Lake Valley elk hunt account described a "404 pound" (184 kg) elk but differentiated it from deer and pronghorn (*Antilocapra americana*). Further, the specific description of antlers "four feet in length" are unlikely to describe head ornaments for any ungulate other than elk [60]. Similarly, the 1847 Ingersoll description also differentiates deer and elk. Further, Ingersoll's account provides detailed miles traveled from the Sierra Nevada crest, likely Donner's Pass, through the South Fork Yuba River watershed then crossing to the Bear River valley en route to "Johnson's",

which would have been Rancho Johnson, a Mexican land grant acquired by William Johnson in 1846 (now Wheatland, Yuba County, California). Ingersoll's elk observation places the animals at about 850 m (2800 feet) elevation along the Bear River in Placer County, 1.6 km (1.0 miles) north of present-day Dutch Flat, California. This observer record is the furthest west in the Sierra Nevada, and at this elevation these likely were not the tule elk subspecies of the Central Valley [5]. The 1874 and 1875 accounts of elk killed at Honey Lake suggest that elk herd sizes may have been substantial, with 15 and 12 elk killed in each hunt, respectively [41, 43, 44]. These numbers were similar to the Round Valley 1874 elk hunt in Plumas County, where thirteen elk were taken [42]. It seems most likely that this Round Valley (there are several in the Sierra Nevada) is the former settlement established in 1863 in Plumas County, as the others are not populated places but rather just geographic features. Of note, the five historical newspaper accounts of elk hunts may have never been recorded if not for the discovery of silver ore in 1859 in Virginia City, Nevada. The prospect of wealth resulted in a population surge and the launching of several regional newspapers only 16 years before our records of elk cease in 1875. Similarly, the location of the 1847 wagon train observer record of elk might have remained inexplicable if not for Ingersoll's punctilious diary entries [40].

The elk hunt historical observer records were supported by two Late Holocene zooarchaeological records in the northwest Great Basin, outside of but adjacent to the northeast Sierra Nevada, one in California and one in western Nevada. This physical evidence of elk in regions close to (just north of) the Sierra Nevada is supported by two reliable historical observer records also outside of but near the eastern and southern borders of the Sierra Nevada, respectively, by the early California explorer and naturalist, Thomas Farnham [30].

Indirect evidentiary records also support the historical presence of elk in the Sierra Nevada, similarly to the direct evidentiary records. Ethnolinguistic and ethnographic evidence from four Native American tribes place elk in the Sierra Nevada and northwestern Great Basin historically. The toponomastic reference to an Elk Point, just south of the hunting record at Zephyr Cove on the southeastern shore of Lake Tahoe, while coincidentally close to the historical elk hunt there, may have an eponymous or other etymology unrelated to elk, the animal.

The paucity of historical observer records and Late Holocene zooarchaeological records are consistent with elk likely occupying the Sierra Nevada at low densities historically. An extensive review of zooarchaeological records posited that elk were never abundant in the Great Basin, finding "20 Holocene-aged Great Basin sites positive for elk, compared to 77 that contained bison (*Bison bison*)—itself never an abundant animal in this region." [28]. The lack of museum records is likely related to the relatively late establishment of museum zoology collections in California, which opened in the late nineteenth century only at the San Francisco California of Sciences and Stanford University's Museum of Natural History–after tule elk had been nearly extirpated [13]. Another factor which may drive the rarity of museum specimens and historical observer records is that the Sierra Nevada was thinly settled and poorly explored by Anglo-Americans until the latter half of the nineteenth century. Finally, Late Holocene zooarchaeological specimens may have been rare, because the region was sparsely populated by Native Americans, so the primary mechanism for introducing elk to caves and rock shelter archaeological sites may have occurred at low frequency [28]. These factors may have also led to a dearth of historical records of moose (*Alces alces*) in Nevada, another deer family member which has now colonized northeastern portions of the state with dispersals noted in the 1950s leading to an established permanent population over the last decade, although this hypothesis has not been verified [61].

Although we present evidence that suggests historical elk occupation of the Sierra Nevada, it is difficult to draw inferences regarding the ability of historical habitat conditions to support elk populations. However, it is clear that contemporary habitat conditions are suitable for

supporting elk populations now and into the future. Habitat suitability in the northern Sierra Nevada is demonstrated by intergrade Roosevelt x Rocky Mountain elk herds exhibiting successful, sustained reproduction in the northern Sierra Nevada as far south as Stampede Reservoir [55]. However, because of their genetic admixture, these intergrade elk may be better adapted to higher elevations and novel terrain than their non-intergrade ancestors. This idea needs to be better explored through rigorous genetic analyses, such as next-generation sequencing, to be substantiated. Further northern range expansion of the nascent Lake Isabella herd in Kern County would suggest that the southern Sierra Nevada is suitable Rocky Mountain elk habitat. The recent recolonization of elk's primary predator, gray wolves, in Tulare County, California in the southern Sierra Nevada, may also imply that elk are a missing component of the ecosystem [21].

Because elk are vagile mammals, with reported dispersal distances up to 600 km [62] further range expansions of elk throughout the Sierra Nevada are not only possible, but contemporary, substantiated observations suggest it is likely. Although dispersal distances for Rocky Mountain elk average 42 km or lower [62], range expansion to the central Sierra Nevada may occur in the near future as long-distance dispersals to the central Sierra Nevada have occurred with increasing frequency over the last decade.

In conclusion, although direct evidence suggests elk were never abundant in the Sierra Nevada historically, we find that the direct and indirect evidence, when taken together, indicates that elk likely occupied the northern Sierra Nevada at least as far south as Lake Tahoe, as well as the northwest Great Basin, at least seasonally, in low densities, or both. Long-distance dispersals to the central Sierra Nevada and recent colonization into the southern Sierra Nevada, as well as a historical observer record of elk further south in the San Bernardino Mountains, raise the further question as to whether most of the Sierra Nevada is suitable elk habitat. Our findings should spur searches for additional physical evidence of elk in the Sierra Nevada during the historical period, utilizing novel approaches and technologies such as those that have recently expanded the recognized historical ranges of other species in California [17, 63, 64]. Although we do not currently make recommendations for or against further elk reintroduction to the Sierra Nevada, it is clear that management activities that support and promote the recently established elk populations in Plumas and Sierra Counties may yield numerous ecological and social benefits. Sustained elk occupation in this wildfire-prone region may help in reduction of fire fuel loads [65, 66], create new wildlife viewing and hunting (including indigenous hunting) opportunities [67], and reestablish an important prey base for apex predators and partial or obligate scavengers including California condors (*Gymnogyps californianus*) [68, 69]. However, given the juxtaposition of elk in the southern Sierra Nevada to the Owens Valley tule elk, there may be future genetic consequences and thus management implications to consider, particularly to preserve the genetic integrity of the endemic tule elk.

Human-elk conflict issues including fence damage and forage competition with livestock should be reported to the California Department of Fish and Wildlife's Wildlife Incident Reporting (WIR) system and managed by established principles of adaptive management [70]. Consideration of wildlife overcrossings should be considered for Interstate 80 and other major highways, considering the proximity of the recolonizing elk in the northern Sierra Nevada. Lastly, we recommend further research to better understand and manage the Sierra Nevada's (re)colonizing elk populations, including application of satellite GPS collars to elucidate space use, habitat preferences, landscape connectivity, identification of unoccupied habitats where range expansion by elk may naturally occur, and DNA analysis of untested herds such as in the southern Sierra Nevada.

## Acknowledgments

The authors wish to acknowledge the thoughtful diligence of Robert "Bob" McQuivey, the retired NDOW investigator who in 1997 researched and discovered the historical newspaper and observer accounts of elk hunts in the Sierra Nevada, but whose detailed work remained unpublished for over a quarter century. We also wish to recognize Ian M. McGlory, MLIS, the Processing Archivist at Mathewson-IGT Knowledge Center, University of Nevada, Reno for his assistance locating images of historical newspaper accounts. We thank the CDFW biologists who provided the latest updates on the expanding elk populations in the northern and southern Sierra Nevada. This article was partially supported by CDFW, NDOW, and New Mexico Department of Game and Fish (NMDGF), with US Fish and Wildlife Service (USFWS), and are solely the responsibility of the authors and do not necessarily represent the official views of the CDFW, NDOW, NMDGF, or USFWS.

## Author Contributions

**Conceptualization:** Richard B. Lanman, Thomas J. Batter, Cody J. Mckee.

**Data curation:** Richard B. Lanman, Cody J. Mckee.

**Formal analysis:** Richard B. Lanman.

**Investigation:** Richard B. Lanman, Thomas J. Batter, Cody J. Mckee.

**Methodology:** Richard B. Lanman, Thomas J. Batter, Cody J. Mckee.

**Project administration:** Richard B. Lanman.

**Resources:** Richard B. Lanman, Thomas J. Batter, Cody J. Mckee.

**Validation:** Richard B. Lanman, Thomas J. Batter, Cody J. Mckee.

**Visualization:** Richard B. Lanman, Thomas J. Batter, Cody J. Mckee.

**Writing – original draft:** Richard B. Lanman, Cody J. Mckee.

**Writing – review & editing:** Richard B. Lanman, Thomas J. Batter, Cody J. Mckee.

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
