## [Decision Letter · Decision Letter 0]

10 Jan 2024

PONE-D-23-41818Novel evidence that elk were historically native to the Sierra Nevada, and recent range expansions into the regionPLOS ONE

Dear Dr. Lanman,

Thank you for submitting your manuscript to PLOS ONE. After careful consideration, we feel that it has merit but does not fully meet PLOS ONE’s publication criteria as it currently stands. Therefore, we invite you to submit a revised version of the manuscript that addresses the points raised during the review process.

<ul><li>Please pay particular attention to addressing the following points from the reviewers:LInes 26 and again on lines 66 and 67: Not much evidence presented in the paper to substainiate the claim of elk populations being suppressed by aboriginal hunting . I would think even if native american hunted elk there would be evidence from petroglyphs, or bone fragments in archeology sites. Please address.Lines 190-193: Although the authors say this is in the Sierra Nevada region. The reference seems to be more relegated to Central to Northeast NV, where the Humboldt River begins (Marys River) and terminates (Humboldt Sink) near Lovelock, NV. I don't know that this provides direct or indirect evidence but it does indicate elk may have been present in the region around the 1840's. Please clarify or modify this claim.Lines 198-203: Same general comment as above. Consider modifying the description here and mentioning in the Discussion.Lines 280-284: It is not clear where this observation is noted on Figure 2.Lines 289-290: I'm not sure if this is a reference to all the observations from 2019 to 2022, or just the observations that were corroborated with fecal DNA? Were they at least from different sexes of elk? That would lend more plausibility of there being a breeding population in the Sierra Nevada. Please clarify

Minor critique: some language could be tightened throughout. For example, line 22. Consider rewording to be more concise: "were previously considered non-native to the Sierra Nevada" perhaps.3`==============================

We look forward to receiving your revised manuscript.

Kind regards,

Mark Zabel

Academic Editor

PLOS ONE

Journal Requirements:

2. In your manuscript, please provide additional information regarding the specimens used in your study. Ensure that you have reported human remain specimen numbers and complete repository information, including museum name and geographic location.

For more information on PLOS ONE's requirements for paleontology and archeology research, see https://journals.plos.org/plosone/s/submission-guidelines#loc-paleontology-and-archaeology-research.

3. We note that Figures 1 & 2 in your submission contain [map/satellite] images which may be copyrighted. All PLOS content is published under the Creative Commons Attribution License (CC BY 4.0), which means that the manuscript, images, and Supporting Information files will be freely available online, and any third party is permitted to access, download, copy, distribute, and use these materials in any way, even commercially, with proper attribution. For these reasons, we cannot publish previously copyrighted maps or satellite images created using proprietary data, such as Google software (Google Maps, Street View, and Earth). For more information, see our copyright guidelines: http://journals.plos.org/plosone/s/licenses-and-copyright.

 a. You may seek permission from the original copyright holder of Figure(s) [#] to publish the content specifically under the CC BY 4.0 license. 

Reviewers' comments:

Reviewer's Responses to Questions

**Comments to the Author**

1. Is the manuscript technically sound, and do the data support the conclusions?

Reviewer #1: Yes

2. Has the statistical analysis been performed appropriately and rigorously? 

Reviewer #1: N/A

3. Have the authors made all data underlying the findings in their manuscript fully available?

Reviewer #1: No

4. Is the manuscript presented in an intelligible fashion and written in standard English?

Reviewer #1: Yes

5. Review Comments to the Author

Reviewer #1: I believe this manuscript is an important contribution to science and understanding of the history of elk distribution in California and in the Sierra Nevada mountains and surrounding areas. The manuscript is well written and makes a compelling argument that Rocky Mountain elk likely inhabited the Sierra Nevada previously to the 1900's, but due to a lack of museum specimens, zooarchaeological or other records, previous texts and publications may have overlooked their presence. This could have been due to a lack of modern records documenting their existence because they were overexploited or simply vary rare, albeit present, in modern history. I believe the authors present more indirect evidence of this than direct evidence. The historical records could be substantiated by more clear references than were presented here. Furthermore, the observations and zooarchaeological records that are presented as "direct evidence" seem to be more indirect or circumstantial to elk being near the Sierra Nevada mountains or perhaps occasional dispersals to the Sierra Nevada. For example the map on Figure 2 shows the locations of historical records of elk and zooarchaeological sites with elk remains, however only 2 out of 6 records appear to be within the identified Sierra Nevada zone while the other 4 records occur just outside of this zone.

The indirect evidence seems more compelling including the ethnographic and ethnolinguistic information presented such as the newspaper articles describing elk being harvested in the Sierra Nevada as well as local place names and native American names for elk in the region and specifically within the Sierra Nevada.

Additionally, as the title implies there has been a recent influx and establishment of Rocky Mountain elk into portions of the northern Sierra Nevada and several documented recent sightings or telemetry movements elsewhere into the Sierra Nevada indicating it is clearly suitable habitat for them. This seems relevant to modern management of both these novel ungulates and the predator prey relationships they may arise from their expansion. This may be very important for understanding the distribution of other species such as modern Gray wolves that have established and also other large carnivores who prey on these species such as mountain lions, black bears, coyotes, and possibly wolverines that scavenge these carcasses. I believe this provides some more evidence that elk could potentially have occurred in the Sierra Nevada historically, assuming there has not been major habitat changes in the past 150 years, and that they likely can persist there in current conditions.

Overall I would recommend publishing this manuscript with a few slight clarifications to references provided and some minor editorial suggestions.

6. PLOS authors have the option to publish the peer review history of their article (what does this mean?). If published, this will include your full peer review and any attached files.

Reviewer #1: No

---

## [Author Response · Author response to Decision Letter 0]

21 Feb 2024

Thank you for the careful reviews. Please see our Responses document in table form.

---

## [Decision Letter · Decision Letter 1]

20 Mar 2024

Novel evidence that elk were historically native to the Sierra Nevada, and recent range expansions into the region

PONE-D-23-41818R1

Dear Dr. Lanman,

We’re pleased to inform you that your manuscript has been judged scientifically suitable for publication and will be formally accepted for publication once it meets all outstanding technical requirements.

Kind regards,

Mark Zabel

Academic Editor

PLOS ONE

Additional Editor Comments (optional):

Reviewers' comments:

Reviewer's Responses to Questions

**Comments to the Author**

1. If the authors have adequately addressed your comments raised in a previous round of review and you feel that this manuscript is now acceptable for publication, you may indicate that here to bypass the “Comments to the Author” section, enter your conflict of interest statement in the “Confidential to Editor” section, and submit your "Accept" recommendation.

Reviewer #1: All comments have been addressed

2. Is the manuscript technically sound, and do the data support the conclusions?

Reviewer #1: Yes

3. Has the statistical analysis been performed appropriately and rigorously? 

Reviewer #1: Yes

4. Have the authors made all data underlying the findings in their manuscript fully available?

Reviewer #1: Yes

5. Is the manuscript presented in an intelligible fashion and written in standard English?

Reviewer #1: Yes

6. Review Comments to the Author

Reviewer #1: The manuscript is a meaningful contribution to science and authors have made appropriate revisions to address concerns of original paper. I believe they completed a comprehensive search of historical and contemporary records of elk or references of elk in the ethnographic and linguistic literature as well as museum records. I believe the authors make a compelling argument, although not without uncertainty, that elk were likely present in the Sierra Nevada prior to European settlement and during early exploration of Anglo-American explorers.

7. PLOS authors have the option to publish the peer review history of their article (what does this mean?). If published, this will include your full peer review and any attached files.

Reviewer #1: No

---

## [Editor Report · Acceptance letter]

29 Apr 2024

PONE-D-23-41818R1 

PLOS ONE

Dear Dr. Lanman, 

I'm pleased to inform you that your manuscript has been deemed suitable for publication in PLOS ONE. Congratulations! Your manuscript is now being handed over to our production team.

Kind regards, 

on behalf of

Dr. Mark Zabel 

Academic Editor

PLOS ONE